# TOWARDS UNIFORMITY AND ALIGNMENT FOR MULTI-MODAL REPRESENTATION LEARNING

## ABSTRACT

Multimodal representation learning aims to construct a shared embedding space in which heterogeneous modalities are semantically aligned. Despite strong empirical results, InfoNCE-based objectives introduce inherent conflicts that yield *distribution gaps* across modalities. We identify and formally analyze two conflicts in the multimodal regime, both exacerbated as the number of modalities $M$ increases: (i) an *alignment–uniformity* conflict, whereby uniform repulsion undermines positive-pair alignment, and (ii) an *intra-alignment* conflict stemming from the non-collinearity of multi-way positives. To address these issues, we propose a principled decoupling of alignment and uniformity. We then demonstrate a theoretical guarantee that our method mitigates the distribution gap by introducing a global Hölder divergence over multiple modality distributions. We show that our decoupled losses act as efficient proxies for minimizing this cross-modal divergence. Extensive experiments on retrieval and UnCLIP-style generation demonstrate consistent gains. Overall, this work provides a conflict-free recipe and theoretical guidance for multimodal learning that simultaneously supports discriminative and generative use cases without task-specific modules.

## 1 INTRODUCTION

Multimodal representation learning (Ruan et al., 2023; Girdhar et al., 2023) aims to construct a shared embedding space where semantically related signals from different modalities (e.g., image, text, audio, video, speech) are well aligned. A landmark example is CLIP (Radford et al., 2021), which employs an InfoNCE objective to align paired image–text representations by maximizing similarity for positive pairs while pushing negative pairs apart. This framework has since been extended beyond two modalities. For instance, ImageBind (Girdhar et al., 2023), VAST (Chen et al., 2023), and LanguageBind (Zhu et al., 2023) incorporate additional streams into a common space, while GRAM (Cicchetti et al., 2024) generalizes the pairwise cosine similarity to the Gramian volume among multiple modalities in the InfoNCE loss and achieves promising performance.

Despite notable successes, InfoNCE-based methods exhibit inherent conflicts that induce distribution gaps (Liang et al., 2022; Shi et al., 2023; Yin et al., 2025). Consequently, UnCLIP-type generative models (e.g., DALL-E 2 (Ramesh et al., 2022) and Kandinsky (Razzhigaev et al., 2023)) add a diffusion module to transform CLIP embeddings. Prior work (Yin et al., 2025) shows that, in vision–language learning, this gap arises from a conflict between *uniformity* and *alignment* (Wang & Isola, 2020): the uniformity term spreads embeddings on the unit hypersphere, whereas the alignment term pulls positive (multimodal) pairs together. However, existing analyses are limited to two modalities and lack a principled extension to the multimodal setting. This leaves a research question: how can we learn representations such that their distributions are well-aligned across modalities (beneficial for generation) while maintaining separability (beneficial for retrieval)? Addressing this requires a comprehensive analysis of the internal conflicts in multimodal learning. Yet the geometry of multimodal representation spaces is more complex and heterogeneous, making it challenging to quantify these conflicts and their scaling with the number of modalities.

**Contributions.** In this work, we take a step forward by systematically analyzing and addressing the inherent conflicts in multimodal InfoNCE that give rise to modality and distributional gaps. Our contributions are four-fold and are highlighted below.

First, we provide a comprehensive theoretical analysis of the InfoNCE objective in multimodal settings, i.e., modality count $M \geq 3$. We theoretically formalize two distinct conflicts: (1) an alignment-uniformity conflict ($\zeta_a$), where uniformity forces oppose alignment, exacerbating distributional gaps across modalities (see Fig. 1a and Corollary 1 in Sec. 2), and (2) an intra-alignment conflict ($\chi_a$), driven by non-collinear positive embeddings across modalities, which widens the modality gap as the number $M$ of modalities increases (see Fig. 1b and Corollary 2 in Sec. 2). Together, these conflicts explain why multimodal InfoNCE struggles to scale: the same objective that enforces global uniformity undermines the alignment of positive pairs, especially as the modality count $M$ grows.

Second, to resolve these issues, we propose a principled decoupling of alignment and uniformity. We enforce *intra-modality uniformity* within each modality's samples, ensuring uniform coverage on the unit hypersphere and preventing representation collapse. In parallel, we introduce an *anchor-based alignment* strategy that aligns embeddings of the same sample across modalities with respect to a designated anchor. This explicitly avoids non-collinearity among positive pairs, thereby closing modality gaps without introducing competing forces. Following the **uni**formity and **align**ment principle, we name our method as *UniAlign*.

Third, beyond this geometric intuition, we provide a theoretical guarantee that our method minimizes the distribution gap. Specifically, we introduce a global Hölder divergence applicable to an arbitrary number of modality distributions. We then connect our decoupled losses to this divergence, showing that the intra-modality uniformity and anchor-based alignment terms act as efficient computational proxies for minimizing it, thereby providing formal theoretical justification.

Extensive experiments demonstrate the effectiveness of our approach. Our framework consistently outperforms InfoNCE-based baselines in representation quality, retrieval accuracy, and generation fidelity. Without additional task-specific modules, the learned embeddings support both discriminative (cross-modal retrieval) and generative (UnCLIP-style conditional generation) tasks, yielding around 2 R@1 gains and 10–40 lower FID, respectively. These results confirm that our decoupled principle not only resolves the modality and distributional gaps introduced by InfoNCE, but also provides a scalable recipe for robust and versatile multimodal learning.

## 2 MOTIVATION: CONFLICT ANALYSIS IN MULTIMODAL LEARNING

In this section, we first revisit the previous analysis of the InfoNCE objective for two modalities (vision and language). Then, we present a general and principled analysis for multimodal learning.

### 2.1 UNIFORMITY AND ALIGNMENT CONFLICT OF INFONCE.

Let $M$ be the number of modalities and $B$ the batch size. For sample index $i \in \{1, \ldots, B\}$ and modality $m \in \{1, \ldots, M\}$, denote the $\ell_2$-normalized embedding by $\mathbf{Z}^{(m)} = \{\mathbf{z}_i^{(m)}\}_{i=1}^B \in \mathbb{R}^d$. The generalized multi-modal InfoNCE objective (Oord et al., 2018) (sum over all paiers) is

$$\mathcal{L}_{\text{InfoNCE}} = -\frac{1}{\sum_{m \neq n} w_{mn}} \sum_{i=1}^{B} \sum_{m \neq n} w_{mn} \log \frac{\exp\left(\mathbf{z}_i^{(m)\top} \mathbf{z}_i^{(n)}/\tau\right)}{\sum_{k=1}^{B} \exp\left(\mathbf{z}_i^{(m)\top} \mathbf{z}_k^{(n)}/\tau\right)}, \quad (1)$$

where $w_{mn} > 0$ denotes weight and $\tau > 0$ is the temperature. This loss has been extensively used in recent multimodal applications (Girdhar et al., 2023; Guzhov et al., 2022). Then for two modalities (i.e., $M = 2$), InfoNCE can be decomposed into *alignment* and *uniformity* (Wang & Isola, 2020) :

$$\text{Alignment: } \mathbb{E}_{p_{\text{pair}}}\big[\|\mathbf{z}^{(1)} - \mathbf{z}^{(2)}\|_2^2\big], \qquad \text{Uniformity: } \log \mathbb{E}_{p_{data}}\big[\exp(-\|\mathbf{z}^{(1)} - \mathbf{z}^{(2)}\|_2^2/2\tau)\big], \quad (2)$$

where $p_{\text{pair}}$ is the paired data distribution, and $p_{\text{data}}$ is the overall data distribution. Uniformity spreads embeddings over the unit hypersphere, thereby avoiding collapse and promoting semantic coverage, while alignment pulls paired cross-modal representations together to enforce semantic consistency. In vision–language learning, Yin et al. (2025) clearly demonstrate that uniformity *across* modalities ("inter-uniformity") conflicts with the alignment term, resulting in a systemic distributional gap.

However, when extending to more modalities, the analysis is insufficient to present the relationship between conflict degree and the number of modalities, which is important to understand the learning

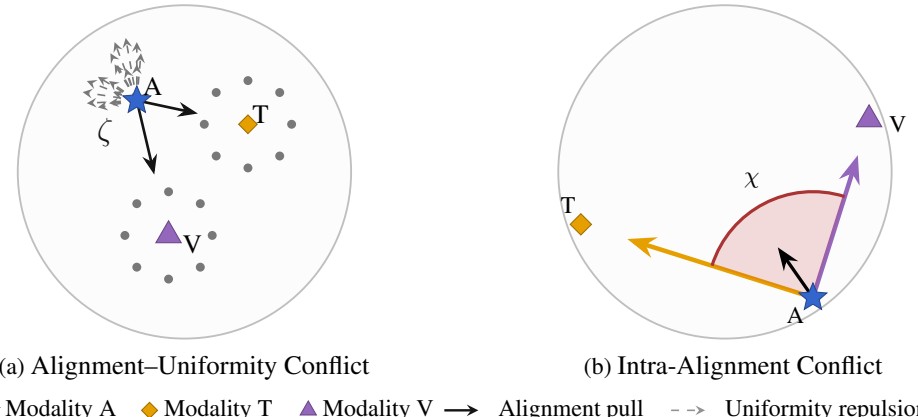

(a) Alignment–Uniformity Conflict    (b) Intra-Alignment Conflict

★ Modality A ◆ Modality T ▲ Modality V ⟶ Alignment pull ⇢ Uniformity repulsion

Figure 1: **Two conflicts of multi-modal InfoNCE.** (a) *Alignment–uniformity:* positives are pulled together yet repelled by the uniformity force; (b) *Intra-alignment:* non-collinear positives induce angular tension. Both grow with $M$.

mechanism of multimodal representation. Due to a much more complex geometry in representation space of multiple modalities , where each modality is influenced by multiple factors, quantifying the conflict in multimodal representation learning is challenging.

## 2.2 SYSTEMIC ANALYSIS CONFLICTS IN MULTIMODAL LEARNING

We first reveal two modes of conflict in multimodal learning with InfoNEC, and then prove that the two conflicts become severe when the number of modalities $M$ increases by Corollary 1 and 2.

To quantify the conflict in multimodal learning, we first choose one modality $\mathbf{z}^{(a)}$ as the anchor, and analyze how it is updated by other modalities from the gradient perspective. Differentiating Eq. (1) with respect to an anchor $\mathbf{z}_i^{(a)}$ exposes a "push–pull" structure. For a single modality pair $(a \rightarrow n)$,

$$\nabla_{\mathbf{z}_i^{(a)}}\mathcal{L} = -\underbrace{\sum_{n \neq a} \frac{w_{an}}{\tau}\mathbf{z}_i^{(n)}}_{\mathbf{V}_a} + \underbrace{\sum_{n \neq a} \frac{w_{an}}{\tau}\sum_{k=1}^{B} p_{ik}^{(an)}\mathbf{z}_k^{(n)}}_{\mathbf{\Phi}_a}, \quad p_{ik}^{(an)} = \frac{\exp(\mathbf{z}_i^{(a)\top}\mathbf{z}_k^{(n)}/\tau)}{\sum_{\ell=1}^{B}\exp(\mathbf{z}_i^{(a)\top}\mathbf{z}_\ell^{(n)}/\tau)}, \quad (3)$$

where $\mathbf{V}_a$ is the resultant force (alignment), and $\mathbf{\Phi}_a$ is the repulsion force (uniformity).

Eq. (3) exposes two levels of conflicts. (i) *Inter-modality alignment–uniformity conflict*: when the uniformity push and alignment pull are directionally aligned, i.e., $\langle \mathbf{V}_a, \mathbf{\Phi}_a \rangle > 0$, the term $-\mathbf{V}_a + \mathbf{\Phi}_a$ cancels in the gradient, yielding a small update for $\mathbf{z}^{(a)}$ (see Fig. 1a). This is induced by *inter-modality* uniformity, leading to a distribution gap. (ii) *Intra-alignment conflict*: the alignment force itself may weaken when the positive targets are not co-linear (see Fig. 1b). Non-collinear or even opposing $\{\mathbf{z}^{(n)}\}_{n\neq a}$ partially cancel in $\mathbf{V}_a = \sum_{n \neq a} a_{an}\mathbf{z}^{(n)}$, producing a weak alignment signal.

**Conflict quantification** $(\zeta_a, \chi_a)$**.** We define alignment-uniformity conflict $\zeta_a \in [-1, 1]$ to measure directional opposition between the alignment and uniformity forces, and introduce the intra-alignment conflict $\chi_a \in [0, 1]$ to quantify cancellation among non-collinear positive pulls within $\mathbf{V}_a$ :

$$\zeta_a \triangleq \cos(\mathbf{V}_a, \mathbf{\Phi}_a) = \mathbf{V}_a^\top \mathbf{\Phi}_a / (\|\mathbf{V}_a\|_2 \|\mathbf{\Phi}_a\|_2). \qquad \chi_a \triangleq 1 - \|\mathbf{V}_a\|_2 / \left(\sum_{n\neq a} w_{an}/\tau\right), \quad (4)$$

A high positive $\zeta_a$ (near 1) indicates severe conflict, which occurs when hard negatives lie in the same direction as positives. $\chi_a$ indicates the magnitude of the vector. A value of $\chi_a = 0$ indicates perfect alignment (no conflict), whereas $\chi_a \rightarrow 1$ indicates severe conflict.

**Assumption 1** (Systematic conflict per-modality )**.** *Let $\hat{\mathbf{V}}_a = \mathbf{V}_a/\|\mathbf{V}_a\|$ denote the unit alignment direction for anchor $a$. For each modality $n \neq a$, the uniformity component admits the decomposition*

$$\mathbf{\Phi}_a^{(n)} = c_n \hat{\mathbf{V}}_a + \varepsilon_n, \qquad (5)$$

*where $c_n \triangleq \langle \mathbf{\Phi}_a^{(n)}, \hat{\mathbf{V}}_a \rangle$ quantifies the magnitude of systematic conflict from modality $n$ in this direction, and satisfies $c_n \geq c_0$ for some positive constant $c_0$ The residuals $\{\varepsilon_n\}_n$ are zero-mean, mutually independent, and have bounded covariance.*

For each modality, non-matching yet semantically similar samples ("hard negatives") exert a weak but systematic pull in the same direction as the true cross-modal alignment; this shared component is modeled by $c_n \hat{\mathbf{V}}_a$. Negatives that are not semantically similar ("easy negatives") are approximately isotropic with respect to $\hat{\mathbf{V}}_a$ and thus have (nearly) zero expected projection; their effects are absorbed into the residual term. Residual variation due to batch composition, data augmentations, and encoder stochasticity is modeled as zero-mean, bounded perturbations $\boldsymbol{\varepsilon}_n$ that are approximately independent across modalities. As the $M$ increases, the systematic components add coherently while the residuals (including easy negatives) average out, leading to the accumulation of alignment–uniformity conflict.

**Corollary 1** (Alignment–Uniformity Conflict). *Let $\boldsymbol{\Phi}_a = \sum_{n \neq a} \boldsymbol{\Phi}_a^{(n)}$ be the total uniformity force on anchor $a$, and define $\zeta_a = \cos(\mathbf{V}_a, \boldsymbol{\Phi}_a)$. Under Assumption 1, the alignment–uniformity conflict converges to its maximum as the number of modalities $M$ increases:*

$$\mathbb{E}[\zeta_a] \;=\; \mathbb{E}\big[\cos(\mathbf{V}_a, \boldsymbol{\Phi}_a)\big] \;\longrightarrow\; 1 \qquad as\ M \to \infty. \tag{6}$$

See its proof in Appendix A. This shows that even if the conflict from each modality ($c_n$) is small, their systematic accumulation inevitably causes the total, observable conflict ($\zeta_a$) to become severe.

**Corollary 2** (Intra-alignment Conflict). *The expected intra-alignment conflict, $\mathbb{E}[\chi_a]$, is governed by $M$ and the average pairwise alignment $\bar{\mu} = \mathbb{E}[\mathbf{z}_i^{(m)\top} \mathbf{z}_i^{(n)}] \in [0, 1]$ for $m \neq n$ between modalities:*

$$\mathbb{E}[\chi_a] \;\geq\; 1 - \sqrt{(1 + (M-2)\bar{\mu})/(M-1)}. \tag{7}$$

*For imperfect alignment ($\bar{\mu} < 1$), the conflict increases with the number of modalities $M$ and admits a non-zero asymptotic lower bound:*

$$\liminf_{M \to \infty} \mathbb{E}[\chi_a] \;\geq\; 1 - \sqrt{\bar{\mu}}. \tag{8}$$

See its proof in Appendix B. Corollary 2 shows that the internal conflict of the alignment force gets severe with more modalities, resulting in ineffective alignment.

By combining Corollary 1 and 2, one can conclude that the standard multi-modal InfoNCE objective is fraught with a two-level conflict system: an intra-alignment conflict, and a classic alignment-uniformity conflict. Such conflicts result in a distinct distributional modality gap. This motivates the exploration of alternative frameworks that **decouple** these objectives, by optimizing uniformity separately and employing a more direct, conflict-free alignment mechanism.

## 3 METHODOLOGY

In Section 2, our analysis has identified two fundamental conflicts that impede multi-modal contrastive learning: the intra-alignment conflict ($\chi$), and the alignment-uniformity conflict ($\zeta$). To circumvent these issues, we propose a generic principle to decouple the learning objectives. Then, we show that our principle essentially minimizes the global distribution gap with a theoretical guarantee.

### 3.1 GENERAL PRINCIPLE FOR MULTIMODAL LEARNING

> **A general principle for multimodal learning**
>
> As the *alignment-uniformity* and intra-alignment conflicts are the root for the modality/distribution gap, a generic principle is to avoid these conflicts from the uniformity and alignment perspectives:
>
> ① **Intra-modality uniformity:** Promote uniform coverage of each modality $m$ on the unit hypersphere $\mathbb{S}^{d-1}$ *within that modality only*. This preserves separability and avoids collapse while *not* inducing inter-modality uniformity forces that oppose alignment.
>
> ② **Conflict-free alignment:** Explicitly or implicitly maximize the consensus magnitude to avoid the non-collinearity problem between positive pairs.
>
> ① avoids the cross-modality uniformity conflict but still pushes the embeddings uniformly spreading in a unit hypersphere. ② avoids the non-collinear positive pulls in the consensus vector.

Following this generic principle, we present our design for the uniformity and alignment terms in Euclidean space. See a summarization of alternatives for each component in Table 4 of Appendix D.

**Uniformity.** To promote uniformity of multimodal representations and mitigate inter-modality conflict, we adopt an *intra-modality* uniformity term to prevent collapse. Let $\mathbf{Z}^{(m)} = \{\mathbf{z}_i^{(m)}\}_{i=1}^B \subset \mathbb{R}^d$ denote a batch of unit-normalized embeddings from modality $m$. Our intra-modality uniformity is

$$U(\mathbf{Z}^{(m)}) = \frac{1}{B} \sum_{i=1}^B \log\left(\frac{1}{B-1} \sum_{j \neq i} \kappa(\mathbf{z}_i^{(m)}, \mathbf{z}_j^{(m)})\right), \quad \kappa(\mathbf{z}_i, \mathbf{z}_j) = \exp\left(-\frac{\|\mathbf{z}_i - \mathbf{z}_j\|_2^2}{2\tau^2}\right), \quad (9)$$

where $\tau > 0$ is the temperature and $\kappa$ is a Gaussian kernel. The sample-wise gradient satisfies $\nabla_{\mathbf{z}_i^{(m)}} U = -\frac{1}{\tau^2} \sum_{j \neq i} p_{ij} (\mathbf{z}_i^{(m)} - \mathbf{z}_j^{(m)})$, $p_{ij} = \frac{\exp(-\|\mathbf{z}_i^{(m)} - \mathbf{z}_j^{(m)}\|_2^2/(2\tau^2))}{\sum_{\ell \neq i} \exp(-\|\mathbf{z}_i^{(m)} - \mathbf{z}_\ell^{(m)}\|_2^2/(2\tau^2))}$, so the softmax weights $p_{ij}$ decay exponentially with distance, effectively suppressing far-away contributions. Consequently, the uniformity term concentrates gradients on nearby hard negatives (semantically similar samples) while leaving distant points largely unaffected, improving local uniformity and preventing collapse. Note that, as the $U(\mathbf{Z}^{(m)})$ is defined for each modality separately, which is different from the $\Phi_a$ term in Eq.( 1), the conflict $\zeta_a$ is avoided.

**Conflict-free alignment.** To address the intra-alignment conflict ($\chi$) inherent in standard multimodal ($M \geq 3$) contrastive objectives, we propose a two-level alignment scheme: (i) an anchor-based alignment and (ii) a volume-based alignment. These either avoid non-collinearity (i) or explicitly promote collinearity among positive pairs (ii).

We choose one modality $\mathbf{Z}^{(a)}$ as the anchor. The anchor modality is optimized only with the uniformity objective, providing a uniform template for the remaining modalities. For the rest, each sample has a single alignment direction to the anchor, avoiding the *non-collinearity* conflict. Hence, we define the alignment loss by the mean squared Euclidean distance over positive pairs:

$$L_{\text{align}} = \frac{1}{B(M-1)} \sum_{i=1}^B \sum_{n \neq a} \|\mathbf{z}_i^{(a)} - \mathbf{z}_i^{(n)}\|_2^2. \quad (10)$$

**Volume-based complement.** The above anchor formulation is a straightforward demonstration. Considering the geometry of multimodal embeddings, both uniformity and alignment can be further strengthened. To this end, we introduce *volume-based* counterparts that operate at the tuple level.

For the volume-based uniformity, we treat each multimodal tuple $\{\mathbf{Z}^{(1)}, \ldots, \mathbf{Z}^{(M)}\}$ (e.g., text, vision, audio) as a single sample via its weighted centroid $\mathbf{C} = \{\mathbf{c}_i\}_{i=1}^B$:

$$\mathbf{c}_i = \sum_{m=1}^M w_m \mathbf{z}_i^{(m)} / \left\| \sum_{m=1}^M w_m \mathbf{z}_i^{(m)} \right\|_2, \qquad w_m \geq 0, \quad \sum_{m=1}^M w_m = 1. \quad (11)$$

We then apply the same uniformity objective to the centroids, $U(\mathbf{C})$, using uniform weights $w_m = 1/M$ by default. This encourages tuple-level dispersion and improves separability for retrieval. Similarly, for alignment, we directly penalize the volume spanned by the modality vectors, which maximizes collinearity among modality embeddings. For each sample $i$, let $\mathbf{G}_i \in \mathbb{R}^{M \times M}$ be the Gram matrix (Cicchetti et al., 2024) with $[\mathbf{G}_i]_{mn} = \langle \mathbf{z}_i^{(m)}, \mathbf{z}_i^{(n)} \rangle$. The simplex volume is proportional to $\sqrt{\det \mathbf{G}_i}$; collinear vectors have zero volume. $\det$ denotes the matrix determinant. Minimizing this term complements the anchor-based alignment by explicitly encouraging collinearity across modalities. Hence, the volume-based complement is given:

$$L_{\text{vol}} = U(\mathbf{C}) + \frac{1}{B} \sum_{i=1}^B (\sqrt{\det \mathbf{G}(\mathbf{z}_i^{(1)}, \ldots, \mathbf{z}_i^{(M)})}. \quad (12)$$

**Overall objective.** The final objective with hyperparameters, $\lambda_{\text{uni}}, \lambda_{\text{align}}, \lambda_{\text{vol}} \geq 0$, is defined by:

$$\mathcal{L} = \lambda_{\text{uni}} \sum_{m=1}^M U(Z^{(m)}) + \lambda_{\text{align}} L_{\text{align}} + \lambda_{\text{vol}} L_{\text{vol}}. \quad (13)$$

**Extension to the unit hypersphere space and geometric constraint.** The above demonstration is one option following the principle. Our principle is generic and can be extended to different variations when considering different representation spaces and geometry properties. For example, as both our uniformity and original InfoNCE loss want embeddings uniformly spreading on the unit hypersphere

$S^{d-1}$, a straightforward design is to use representations often based on geodesic distance in the hypersphere space instead of the Euclidean space. To this end, the geodesic distance:

$$d_{\mathbb{S}}(\mathbf{z}^i, \mathbf{z}^j) = \arccos\big(\langle \mathbf{z}^i, \mathbf{z}^j \rangle\big), \quad k_{\mathbb{S}}(\mathbf{z}^i, \mathbf{z}^j; \tau) = \exp\Big(-\frac{d_{\mathbb{S}}(\mathbf{z}^i, \mathbf{z}^j)^2}{2\,\tau^2}\Big). \qquad \|\mathbf{z}^i\|_2 = \|\mathbf{z}^j\|_2 = 1. \tag{14}$$

Thus, our principle is generic and flexible across design choices. We present instantiations in both Euclidean and manifold settings in Table 4 of Appendix D.

## 3.2 THEORETICAL ANALYSIS FROM DIVERGENCE PERSPECTIVE

In the previous section, we introduced our objective based on the proposed principle. A natural question is whether this objective is theoretically guaranteed to reduce the distribution gap across modalities. Here, we show that optimizing intra-modality uniformity and cross-modality alignment minimizes a global distribution divergence, thereby mitigating the cross-modal (distribution) gap.

Classical divergences (Jenssen et al., 2006; Shlens, 2014) are typically defined between two distributions, but our setting involves $M$ modalities. We therefore introduce a new *global Hölder divergence* to jointly measure the discrepancy among all modality distributions. Let $\{p_m(z)\}_{m=1}^M$ denote the densities of the $M$ modalities. By Hölder's inequality, they satisfy

$$\left| \int \prod_{m=1}^M p_m(z)\, dz \right|^M \leq \prod_{m=1}^M \int |p_m(z)|^M dz, \tag{15}$$

and takes equality if and only if $p_1 = \cdots = p_M$.. This inequality motivates the definition of the global Hölder divergence as the log of ratio between the LHS and RHS or Eq. 15:

$$D_{\text{Hölder}} = -\log \frac{\int \prod_{m=1}^M p_m(\mathbf{z})\, d\mathbf{z}}{\left(\prod_{m=1}^M \int |p_m(\mathbf{z})|^M d\mathbf{z}\right)^{\frac{1}{M}}} = \underbrace{\frac{1}{M} \sum_{m=1}^M \log \int |p_m(\mathbf{z})|^M d\mathbf{z}}_{\text{Uniformity Term}} - \underbrace{\log \int \prod_{m=1}^M p_m(\mathbf{z})\, d\mathbf{z}}_{\text{Alignment Term}}. \tag{16}$$

We empirically estimate this global divergence in a non-parametric way via the kernel density estimator (KDE) with a Gaussian kernel $\kappa(\mathbf{z}_i, \mathbf{z}_j) = \exp\big(-\|\mathbf{z}_i - \mathbf{z}_j\|_2^2/(2\tau^2)\big)$. Our intra-modality uniformity loss $U(\mathbf{Z}^{(m)})$ acts as a computational proxy for the uniformity component of the divergence by enforcing repulsion and promoting per-distribution entropy. Concurrently, our instance-wise alignment loss provides a tractable objective for the alignment component by targeting its low-temperature limit ($\tau \to 0$), where the goal of distributional overlap becomes instance matching.

Thus, our principle is theoretically guaranteed and offers a principled and computationally efficient method for minimizing the global Hölder divergence in Eq. (16). Full KDE derivation for the empirical estimator of the global Hölder divergence is provided in Appendix C.

## 4 RELATED WORK

CLIP (Radford et al., 2021) pioneered aligning two modalities (vision and language) using the InfoNCE objective (Oord et al., 2018). It has enabled substantial progress in image–text retrieval (Jang et al., 2024; Koukounas et al., 2024; Huang et al., 2024) and text-to-image (T2I) generation (Ramesh et al., 2022; Rombach et al., 2022). CLIP-style contrastive objectives have since been applied to additional modality pairs, including audio–text (Elizalde et al., 2023; Wu et al., 2023) and point cloud–text (Zhang et al., 2022). Beyond pairs, recent work such as CMRC (Wang et al., 2023b), CLIP4VLA (Ruan et al., 2023), ImageBind (Girdhar et al., 2023), and LanguageBind (Zhu et al., 2023) extends CLIP by introducing more modalities (e.g., video, audio, depth, IMU) into a unified space using pairwise InfoNCE objectives. In parallel, VAST (Chen et al., 2023), mPLUG-2 (Xu et al., 2023), and InternVideo2 (Wang et al., 2024) advance the state of the art through large-scale training and architectural refinements. Complementing these trends, GRAM (Cicchetti et al., 2024) introduces the cross-modality Gram matrix to replace pairwise cosine similarity in InfoNCE with a volume score given by the modality Gram matrix to better handle multimodal alignment.

Despite the success of these methods, embeddings from different modalities still exhibit distinct *distribution gaps* (Fig. 2), largely attributable to the InfoNCE objective. Prior studies (Zhou et al.,

Table 1: **Zero-shot multimodal text-to-video (T2V) and video-to-text (V2T) retrieval results (Recall@1).** Our method, UniAlign, consistently outperforms baselines in most tasks.

| Method | Modality | MSR-VTT | | DiDeMo | | ActivityNet | | Average | |
|---|---|---|---|---|---|---|---|---|---|
| | | T2V | V2T | T2V | V2T | T2V | V2T | T2V | V2T |
| UMT (Liu et al., 2022) | T–V | 33.3 | – | 34.0 | – | 31.9 | – | 33.1 | – |
| OmniVL (Wang et al., 2022a) | T–V | 34.6 | – | 33.3 | – | – | – | 34.0 | – |
| UMT-L (Li et al., 2023) | T–V | 40.7 | 37.1 | 48.6 | 49.9 | 41.9 | 39.4 | 43.7 | 42.1 |
| TVTSv2 (Zeng et al., 2023) | T–V | 38.2 | – | 34.6 | – | – | – | 36.4 | – |
| ViCLIP (Wang et al., 2023a) | T–V | 42.4 | 41.3 | 18.4 | 27.9 | 15.1 | 24.0 | 25.3 | 31.1 |
| VideoCoCa (Yan et al., 2022) | T–V | 34.3 | 64.7 | – | – | 34.5 | 33.0 | 34.4 | 48.9 |
| Norton (Lin et al., 2024) | T–V | 10.7 | – | – | – | – | – | 10.7 | – |
| ImageBind (Girdhar et al., 2023) | T–V | 36.8 | – | – | – | – | – | 36.8 | – |
| InternVideo-L (Wang et al., 2022b) | T–V | 40.7 | 39.6 | 31.5 | 33.5 | 30.7 | 31.4 | 34.3 | 34.8 |
| HiTeA (Ye et al., 2023) | T–V | 34.4 | – | 43.2 | – | – | – | 38.8 | – |
| mPLUG-2 (Xu et al., 2023) | T–V | 47.1 | – | 45.7 | – | – | – | 46.4 | – |
| VideoPrism-b (Zhao et al., 2024) | T–V | 51.4 | 50.2 | – | – | 49.6 | 47.9 | 50.5 | 49.1 |
| LanguageBind (Zhu et al., 2023) | T–V | 44.8 | 40.9 | 39.9 | 39.8 | 41.0 | 39.1 | 41.9 | 39.9 |
| VAST (Chen et al., 2023) | T–VA | 49.3 | 43.7 | 49.5 | 48.2 | 51.4 | 46.8 | 50.1 | 46.2 |
| GRAM (Cicchetti et al., 2024) | T–VA | 54.2 | 50.5 | 54.2 | **52.2** | 59.0 | 50.4 | 55.8 | 51.0 |
| UniAlign (Ours) | T–VA | **58.7** | **54.6** | **58.2** | 51.6 | **59.4** | **51.7** | **58.8** | **52.6** |

2023; Liang et al., 2022; Shi et al., 2023) have reported this phenomenon in vision–language learning: Liang et al. (2022) observe that the InfoNCE objective can encourage modality gaps, while Yin et al. (2025) provide a theoretical account showing that uniformity and alignment (Wang & Isola, 2020) conflict, inducing persistent distributional discrepancies. However, these analyses are restricted to the bimodal case; a principled understanding of the conflict mechanisms in the *multimodal* regime remains lacking, partly due to the geometric complexity of shared representation spaces. In this work, we systematically analyze these conflicts for general multimodal learning and, based on this analysis, propose a generic principle for multimodal representation learning.

# 5 EXPERIMENTS

We evaluate our method (UniAlign) from two aspects: (i) embedding separability and (ii) the distributional modality gap. For (i), we assess the video retrieval performance (Section 5.1); for (ii), we perform cross-modal generation using fixed image decoders (Section 5.2). More implementation details and experimental results can be found in Appendix F.

## 5.1 VIDEO RETRIEVAL

**Experimental setting.** Following GRAM (Cicchetti et al., 2024), we train on VAST150K (Chen et al., 2023) and evaluate zero-shot video retrieval on three standard benchmarks: MSR-VTT (Xu et al., 2016), DiDeMo (Anne Hendricks et al., 2017), and ActivityNet (Caba Heilbron et al., 2015). For fair comparison, we keep modality encoders identical to VAST/GRAM: BERT-B for text, BEATs (Chen et al., 2022) for audio, and EVA-CLIP ViT-G (Sun et al., 2023) for video. We report zero-shot Recall@1 (R@1) for both text-to-video (T2V) and video-to-text (V2T). For joint multimodal retrieval, where text queries retrieve the most compatible video + audio tuple (T-VA) and vice versa, we follow GRAM and use the *volume* score, i.e., the determinant of the cross-modal Gram matrix.

**Experimental results.** Table 1 shows that our method consistently outperforms the baselines across most tasks. All methods (VAST, GRAM, and ours) use the *same* backbone architecture. Initialized from the VAST pretrained weights, our approach improves VAST by approximately 8 R@1 on T2V and 6 R@1 on V2T, demonstrating the effectiveness of our principle-instantiated objective. Notably, VAST (pairwise InfoNCE) suffers from *both* the alignment–uniformity and intra-alignment conflicts; GRAM, which uses a Gramian volume score within InfoNCE, mitigates the *intra-alignment* conflict by promoting collinearity (the Gramian volume is minimized when vectors are collinear), but the *alignment–uniformity* conflict remains. By addressing *both* conflicts, our method further improves over GRAM (about +3 R@1 on T2V and +1.6 on V2T), underscoring the benefit of explicitly resolving InfoNCE's internal tensions in the multimodal regime.

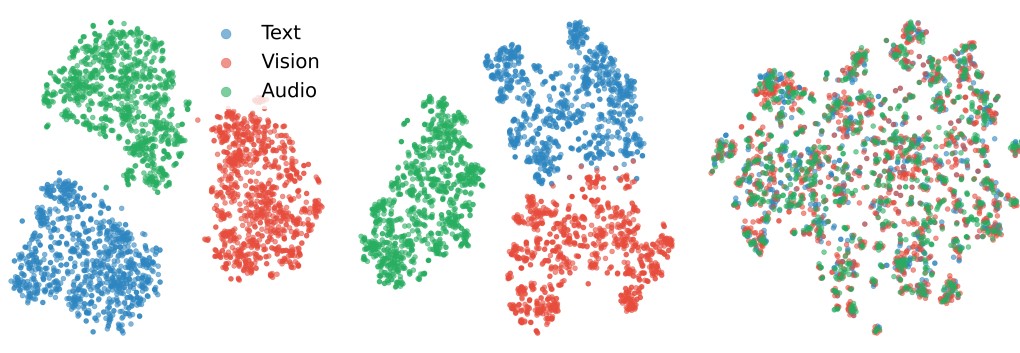

(a) ImageBind (paired InfoNCE).     (b) GRAM (volume InfoNCE).     (c) Ours

Figure 2: **T-SNE visualizations of vision, text, and audio features.** InfoNCE-type of objective results in clear distribution gaps (2a and 2b). Our method mitigates the distribution gap (2c).

**Ablation study.** To understand how the volume-based uniformity $U(C)$ using centroid $C$ and the volume-based alignment ($L_{\mathrm{vol}}$) affect the retrieval performance, we ablate both components to quantify their individual and combined contributions to retrieval performance. We train and evaluate the performance on the MSRVTT dataset. To exclude external factors (e.g., similarity matrix refinement or additional image-text matching refinement) that are commonly used in retrieval, we use the plain cosine

Table 2: **Ablation on $U(C)$ and $L_{\mathrm{vol}}$.**

| $U(C)$ | $L_{\mathrm{vol}}$ | T2V | V2T | Avg. |
|---|---|---|---|---|
| ✗ | ✗ | 36.5 | 36.8 | 36.6 |
| ✗ | ✓ | 38.3 | 36.9 | 37.6 |
| ✓ | ✗ | 37.4 | 39.8 | 38.6 |
| ✓ | ✓ | 40.2 | 43.4 | 41.8 |

similarity-based retrieval for this ablation. As shown in Table 2, both the volume-based uniformity and alignment can effectively increase embedding separability, and thus improve the retrieval performance on MSRVTT. More ablation studies can be found in the Appendix F.2.

## 5.2 CROSS-MODAL GENERATION

To evaluate the distributional modality gap, we use a simple proxy: if multiple modalities are well aligned to a common distribution, embeddings from non-image modalities (e.g., audio or text) should be usable by an image generator trained on image embeddings. In that case, cross-modal generation quality directly reflects the degree of cross-modal alignment. To this end, we employ UnCLIP-type generators, which consist of a separate image generator trained on CLIP image embeddings.

**Dataset.** We use the VGGSound dataset (Chen et al., 2020) to evaluate the generation performance. VGGSound is an audio-visual correspondent, allowing us to build a semantically aligned vision-audio-text triplet. VGGSound has around 200K video clips, annotated with 309 sound classes. The dataset does not provide the video caption. Hence, we use the captioner provided by VAST (Chen et al., 2023) to generate video captions. Then, we use 1024 videos for testing, and the rest for training.

**Experimental setting.** We map all modalities into a shared image-anchored embedding space and evaluate two encoder–decoder configurations compatible with UnCLIP-style generators. (i) *ViT-H*: a CLIP ViT-H/14 image-text encoders paired with the ImageBind audio encoder, compatible with the Stable UnCLIP decoder (Ramesh et al., 2022). (ii) *ViT-bigG*: a CLIP ViT-bigG-14 text/vision model combined with the BEATs audio encoder (Chen et al., 2022), compatible with the Kandinsky decoder (Razzhigaev et al., 2023). We re-train GRAM and use the released ImageBind weights pretrained on large-scale data for comparison. We evaluate text-to-image (T2I), audio-to-image (A2I), and modality interpolation with Fréchet Inception Distance (FID) (Heusel et al., 2017).

**T-SNE visualization.** We first visualize the joint embedding space using 2D t-SNE (Fig. 2) to illustrate modality gaps under different training objectives. We extract text, vision, and audio embeddings from the VGGSound test set, $\ell_2$-normalize them, and compute t-SNE. As shown in Fig. 2, training with an InfoNCE-type objective yields clearly separated, modality-specific clusters (i.e., modality gap), whereas our method produces substantially tighter cross-modal co-location, indicating a smaller modality gap.

**Results.** We compare against GRAM and ImageBind using both *Kandinsky* and *Stable UnCLIP* decoders. When fed image embeddings, these decoders achieve FID 32.99 and 34.61, respectively,

Table 3: **Cross-modal generation with different decoders.** We report FID (↓). Kandinsky and Stable UnCLIP, evaluated in self-reconstruction by feeding image embeddings to the decoder (marked ∗), serve as upper-bound references. Our method consistently outperforms both baselines.

| Decoder | Method | T2I ↓ | A2I ↓ | (T+A)→I ↓ | Avg. ↓ |
|---|---|---|---|---|---|
| Kandinsky | Kandinsky | - | - | - | 32.99* |
| | GRAM | 62.11 | 106.97 | 92.63 | 87.23 |
| | Ours (Geodesic) | **45.35** | **50.75** | **48.19** | **48.09** |
| | Ours (Euclidean) | **42.72** | **50.51** | **46.56** | **46.60** |
| Stable UnCLIP | Stable UnCLIP | - | - | - | 34.61* |
| | ImageBind | 50.17 | 53.59 | 46.81 | 50.19 |
| | GRAM | 45.53 | 55.40 | 47.15 | 49.36 |
| | Ours (Geodesic) | **39.88** | **40.16** | **40.80** | **40.23** |
| | Ours (Euclidean) | **39.63** | **39.95** | **41.03** | **40.20** |

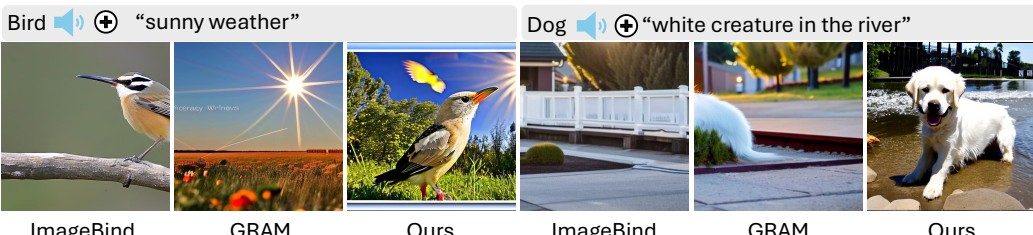

Figure 3: **Modality-interpolation generation results (T+A) → I.** When interpolating between text and audio representations, our method has a better ability to fuse the semantic information across modalities, leading to better generation.

providing dataset-specific upper bounds (decoder self-reconstruction). Our method yields substantial gains in cross-modal generation for both text-to-image (T2I) and audio-to-image (A2I) relative to InfoNCE-trained baselines. As shown in Table 3, with *Kandinsky* we significantly outperform GRAM; with *Stable UnCLIP* we surpass ImageBind and GRAM by around 10 FID. These improvements are consistent across architectures and decoders, indicating the robustness of our objective. The performance further suggests that the learned cross-modal representations are tightly aligned (i.e., the modality gap is small). We also evaluate a geodesic-kernel variant of our method, which performs on par with the Euclidean version, indicating robustness to the choice of geometry and supporting the generality of our principle. Additional qualitative results are provided in Appendix F.

**Modality interpolation.** If modalities are aligned to a common distribution, a straightforward application is *embedding interpolation*, which blends information from different modalities directly in the shared space for image synthesis (as opposed to conditioning a generator via cross-attention from a single modality). Prior work has primarily demonstrated this for vision–language with DALL·E 2 (Ramesh et al., 2022). Here, we interpolate modality embeddings (e.g., $(T+A)/2$ and generate images with Kandinsky and Stable UnCLIP decoders. Our method outperforms baselines both quantitatively (Table. 3) (lower FID) and qualitatively (Fig. 3), indicating an improved ability to fuse complementary semantics across modalities. We attribute these gains to the reduced cross-modal distribution gap and the resulting smoothness of the shared embedding manifold.

## 6 CONCLUSION

We introduced a conflict-aware principle for multimodal representation learning that decouples *uniformity* from *alignment*, overcoming key limitations of InfoNCE when modality number $M \geq 3$. By promoting intra-modality uniformity and anchoring positive alignment, our method directly reduces cross-modal distribution gaps. A divergence-based analysis further shows that these objectives serve as tractable estimators for minimizing a global discrepancy, providing theoretical guarantees. Empirically, the learned embeddings achieve strong performance in video retrieval and cross-modal generation with UnCLIP decoders, while t-SNE visualizations confirm improved modality integration. Overall, our approach offers a conflict-free and theoretically grounded framework for unifying discriminative and generative multimodal tasks without task-specific modules.

USE OF LARGE LANGUAGE MODELS (LLMS).

LLMs (e.g., ChatGPT) were only used for minor language polishing. They did not contribute to research ideation, experimental design, or substantive writing.

ETHICS STATEMENT

Our work presents a fundamental theoretical analysis and a generic principle concerning the training dynamics of multimodal models. As our contribution is primarily theoretical, it is agnostic to specific datasets or downstream applications and does not introduce new, direct ethical risks.

REPRODUCIBILITY STATEMENT

We provide the sufficient implementation details in Section 3.1, Section 5, and Appendix F for reproducibility.

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

# A  PROOF OF COROLLARY 1

**Corollary 1** (Alignment–uniformity Conflict). *Let $\mathbf{\Phi}_a = \sum_{n \neq a} \mathbf{\Phi}_a^{(n)}$ be the total uniformity force on anchor $a$. Assume each per-modality component admits the decomposition:*

$$\mathbf{\Phi}_a^{(n)} = c_n \hat{\mathbf{V}}_a + \boldsymbol{\varepsilon}_n.$$

*Here, $\hat{\mathbf{V}}_a \triangleq \mathbf{V}_a / \|\mathbf{V}_a\|$ is the direction of the total alignment force. The scalar $c_n \triangleq \mathbf{\Phi}_a^{(n)} \cdot \hat{\mathbf{V}}_a$ quantifies the magnitude of systematic conflict from modality $n$ in this direction, and satisfies $c_n \geq c_0$ for some positive constant $c_0$. The vector $\boldsymbol{\varepsilon}_n$ is a random perturbation unique to modality $n$, assumed to be zero-mean, mutually independent, and with bounded covariance.*

*Under these assumptions, the overall alignment-uniformity conflict $\zeta_a$ converges to its maximum value as the number of modalities $M$ increases:*

$$\mathbb{E}[\zeta_a] = \mathbb{E}\big[\cos\big(\mathbf{V}_a, \mathbf{\Phi}_a\big)\big] \to 1 \quad as\ M \to \infty. \tag{17}$$

*Proof.* The proof proceeds in three stages. First, we provide a formal justification for the decomposition of the per-modality uniformity force. Second, we derive a precise expression for the conflict metric $\zeta_a$ based on this decomposition. Finally, we analyze the asymptotic behavior of this expression as the number of modalities $M \to \infty$.

**Justification of the Decomposition**  The decomposition of $\mathbf{\Phi}_a^{(n)}$ is a formalization of the geometric principle of orthogonal projection. For any vector $\mathbf{\Phi}_a^{(n)}$ and a given direction defined by the unit vector $\hat{\mathbf{V}}_a$, we can uniquely decompose $\mathbf{\Phi}_a^{(n)}$ into a component parallel to $\hat{\mathbf{V}}_a$ and a component orthogonal to it. The component parallel to $\hat{\mathbf{V}}_a$ is its orthogonal projection, which we define as the **systematic component**:

$$\mathrm{Proj}_{\hat{\mathbf{V}}_a}(\mathbf{\Phi}_a^{(n)}) = (\mathbf{\Phi}_a^{(n)} \cdot \hat{\mathbf{V}}_a)\hat{\mathbf{V}}_a. \tag{18}$$

Intuitively, in each modality, non-paired but semantically similar samples ("hard negatives") exert a weak but systematic pull in the same direction as the true cross-modal target; this shared component is modeled by $c_n \hat{\mathbf{V}}_a$. Residual variation due to batch composition, data augmentations, and encoder stochasticity is captured by zero-mean, bounded perturbations $\boldsymbol{\varepsilon}_n$ that are approximately independent across modalities. As the number of modalities increases, the systematic components add coherently while the residuals average out, leading to the observed accumulation of alignment–uniformity conflict.

**Derivation of the Conflict Metric $\zeta_a$**  The conflict metric $\zeta_a$ is the cosine similarity between $\mathbf{V}_a$ and the total uniformity force $\mathbf{\Phi}_a$:

$$\zeta_a = \cos(\mathbf{V}_a, \mathbf{\Phi}_a) = \frac{\mathbf{V}_a \cdot \mathbf{\Phi}_a}{\|\mathbf{V}_a\|\|\mathbf{\Phi}_a\|} = \frac{\hat{\mathbf{V}}_a \cdot \mathbf{\Phi}_a}{\|\mathbf{\Phi}_a\|}.$$

Let $N = M - 1$. The total uniformity force is $\mathbf{\Phi}_a = \sum_{n=1}^{N} \mathbf{\Phi}_a^{(n)} = (\sum_{n=1}^{N} c_n)\hat{\mathbf{V}}_a + \sum_{n=1}^{N} \boldsymbol{\varepsilon}_n$. Let $S_c = \sum_{n=1}^{N} c_n$ and $\mathbf{S}_\varepsilon = \sum_{n=1}^{N} \boldsymbol{\varepsilon}_n$. Due to orthogonality, the numerator of $\zeta_a$ is $\hat{\mathbf{V}}_a \cdot \mathbf{\Phi}_a = S_c$ and the squared norm of the denominator is $\|\mathbf{\Phi}_a\|^2 = S_c^2 + \|\mathbf{S}_\varepsilon\|^2$. Substituting these back, we obtain a precise expression for $\zeta_a$:

$$\zeta_a = \frac{S_c}{\sqrt{S_c^2 + \|\mathbf{S}_\varepsilon\|^2}} = \frac{1}{\sqrt{1 + \frac{\|\mathbf{S}_\varepsilon\|^2}{S_c^2}}}.$$

**Asymptotic Analysis**  The proof now hinges on showing that the ratio $\frac{\|\mathbf{S}_\varepsilon\|^2}{S_c^2}$ converges to zero as $N \to \infty$. The denominator $S_c^2 = (\sum c_n)^2 \geq (Nc_0)^2$ grows at least quadratically. For the numerator, due to the independence and zero-mean properties of $\{\boldsymbol{\varepsilon}_n\}$, its expected value grows at most linearly:

$$\mathbb{E}[\|\mathbf{S}_\varepsilon\|^2] = \sum_{n=1}^{N} \mathbb{E}[\|\boldsymbol{\varepsilon}_n\|^2] \leq NC_\varepsilon,$$

for some constant $C_\varepsilon < \infty$ implied by the bounded covariance. The ratio of the expected numerator to the lower-bounded denominator is of the order $O(N)/O(N^2) = O(1/N)$, which converges to 0. This implies that the random variable $\frac{\|\mathbf{S}_\varepsilon\|^2}{S_c^2}$ converges to 0 in probability.

By the Continuous Mapping Theorem, $\zeta_a$ converges in probability to 1. As $\zeta_a$ is bounded in $[-1, 1]$, the Bounded Convergence Theorem ensures that convergence in probability to a constant implies convergence in expectation. Therefore:

$$\lim_{M \to \infty} \mathbb{E}[\zeta_a] = 1. \qquad \square$$

## B   PROOF OF COROLLARY 2

**Corollary 2** (Intra-alignment Conflict). *The expected intra-alignment conflict, $\mathbb{E}[\chi_a]$, is governed by $M$ and the average pairwise alignment $\bar{\mu} = \mathbb{E}[\mathbf{z}_i^{(m)\top}\mathbf{z}_i^{(n)}] \in [0, 1]$ for $m \neq n$ between modalities:*

$$\mathbb{E}[\chi_a] \geq 1 - \sqrt{\frac{1 + (M - 2)\bar{\mu}}{M - 1}}. \tag{19}$$

*For imperfect alignment ($\bar{\mu} < 1$), the conflict increases with the number of modalities $M$ and admits a non-zero asymptotic lower bound:*

$$\liminf_{M \to \infty} \mathbb{E}[\chi_a] \geq 1 - \sqrt{\bar{\mu}}.$$

*Proof.* The intra-alignment conflict for an anchor modality $a$ is defined as:

$$\chi_a \triangleq 1 - \frac{\|\mathbf{V}_a\|_2}{\sum_{n \neq a} w_{an}/\tau}$$

where the alignment force is $\mathbf{V}_a = \sum_{n \neq a} \frac{w_{an}}{\tau} \mathbf{z}_i^{(n)}$. To derive the fundamental scaling relationship with the number of modalities $M$, we make a simplifying assumption of uniform weighting, i.e., $w_{an}/\tau = 1$ for all $n \neq a$. Under this assumption,

$$\chi_a = 1 - \frac{\|\mathbf{V}_a\|_2}{M - 1}, \qquad \mathbf{V}_a = \sum_{n \neq a} \mathbf{z}_i^{(n)}.$$

Let $N = M - 1$. Then

$$\|\mathbf{V}_a\|_2^2 = \sum_{n=1}^{N} \sum_{m=1}^{N} \mathbf{z}_i^{(n)\top}\mathbf{z}_i^{(m)} = N + \sum_{n \neq m} \mathbf{z}_i^{(n)\top}\mathbf{z}_i^{(m)}.$$

Taking expectations and using $\bar{\mu} = \mathbb{E}[\mathbf{z}_i^{(n)\top}\mathbf{z}_i^{(m)}]$ for $n \neq m$,

$$\mathbb{E}[\|\mathbf{V}_a\|_2^2] = N + N(N-1)\bar{\mu} = (M-1)(1 + (M-2)\bar{\mu}).$$

By Jensen's inequality (since $\sqrt{\cdot}$ is concave),

$$\mathbb{E}[\|\mathbf{V}_a\|_2] \leq \sqrt{\mathbb{E}[\|\mathbf{V}_a\|_2^2]} = \sqrt{(M-1)(1 + (M-2)\bar{\mu})}.$$

Hence

$$\mathbb{E}[\chi_a] = 1 - \frac{\mathbb{E}\|\mathbf{V}_a\|_2}{M - 1} \geq 1 - \sqrt{\frac{1 + (M - 2)\bar{\mu}}{M - 1}},$$

which proves equation 19. Finally,

$$\lim_{M \to \infty} \sqrt{\frac{1 + (M - 2)\bar{\mu}}{M - 1}} = \sqrt{\bar{\mu}} \quad \Rightarrow \quad \liminf_{M \to \infty} \mathbb{E}[\chi_a] \geq 1 - \sqrt{\bar{\mu}}.$$

$$\square$$

## C  GENERALIZED HÖLDER DIVERGENCE

**KDE estimation of the global Hölder divergence.**    Let $\{p_m(\mathbf{z})\}_{m=1}^M$ be the (unknown) continuous densities of $M$ modalities and

$$
\begin{aligned}
D_{\text{Hölder}} \;&=\; -\log \frac{\int \prod_{m=1}^M p_m(\mathbf{z})\,d\mathbf{z}}{\left(\prod_{m=1}^M \int p_m(\mathbf{z})^M\,d\mathbf{z}\right)^{1/M}} \\
&=\; \frac{1}{M}\sum_{m=1}^M \log\!\left(\int p_m(\mathbf{z})^M\,d\mathbf{z}\right) \;-\; \log\!\left(\int \prod_{m=1}^M p_m(\mathbf{z})\,d\mathbf{z}\right),
\end{aligned}
\tag{20}
$$

which is nonnegative by Hölder's inequality and equals 0 iff the Hölder inequality is tight.

For modality $m$, let $\{\mathbf{z}_k^{(m)}\}_{k=1}^B$ be a batch of embeddings on $\mathbb{R}^d$ and define the kernel density estimator (KDE)

$$
\widehat{p}_m(\mathbf{z}) \;=\; \frac{1}{B}\sum_{k=1}^B K_\tau\big(\mathbf{z}, \mathbf{z}_k^{(m)}\big), \qquad K_\tau(\mathbf{z}, \mathbf{z}') \;=\; \frac{1}{(2\pi\tau^2)^{d/2}}\exp\!\Big(-\frac{\|\mathbf{z}-\mathbf{z}'\|_2^2}{2\tau^2}\Big),
\tag{21}
$$

with bandwidth $\tau > 0$.[1]

Using $\int p_m^M = \mathbb{E}_{Z\sim p_m}\big[p_m(Z)^{M-1}\big]$ and $\int \prod_{m=1}^M p_m = \mathbb{E}_{Z\sim p_1}\big[\prod_{m=2}^M p_m(Z)\big]$, we obtain Monte-Carlo plug-in estimators by sampling from the empirical $p_m$ via $\{\mathbf{z}_j^{(m)}\}$ and evaluating the KDEs:

$$
\int \widehat{p}_m(\mathbf{z})^M\,d\mathbf{z} = \mathbb{E}_{Z\sim\widehat{p}_m}\big[\widehat{p}_m(Z)^{M-1}\big] \;\approx\; \frac{1}{B}\sum_{j=1}^B\left(\frac{1}{B}\sum_{k=1}^B K_\tau\big(\mathbf{z}_j^{(m)}, \mathbf{z}_k^{(m)}\big)\right)^{M-1},
\tag{22}
$$

$$
\int \prod_{m=1}^M \widehat{p}_m(\mathbf{z})\,d\mathbf{z} = \mathbb{E}_{Z\sim\widehat{p}_1}\Big[\prod_{m=2}^M \widehat{p}_m(Z)\Big] \;\approx\; \frac{1}{B}\sum_{j=1}^B \prod_{m=2}^M\left(\frac{1}{B}\sum_{k=1}^B K_\tau\big(\mathbf{z}_j^{(1)}, \mathbf{z}_k^{(m)}\big)\right).
\tag{23}
$$

Equivalently, with the unnormalized Gaussian kernel $\kappa$ (dropping constants), the formulas above match

$$
\int p_m^M \approx \frac{1}{B}\sum_{j=1}^B\Big(\frac{1}{B}\sum_{k=1}^B \kappa\big(\mathbf{z}_j^{(m)}, \mathbf{z}_k^{(m)}\big)\Big)^{M-1}, \qquad \int \prod_{m=1}^M p_m \approx \frac{1}{B}\sum_{j=1}^B \prod_{m=2}^M\Big(\frac{1}{B}\sum_{k=1}^B \kappa\big(\mathbf{z}_j^{(1)}, \mathbf{z}_k^{(m)}\big)\Big).
$$

(i) *Leave-one-out (LOO).* To reduce small-sample bias, one may replace $\frac{1}{B}\sum_{k=1}^B K_\tau(\mathbf{z}_j^{(m)}, \mathbf{z}_k^{(m)})$ by $\frac{1}{B-1}\sum_{k\neq j} K_\tau(\mathbf{z}_j^{(m)}, \mathbf{z}_k^{(m)})$ in equation 22. (ii) *Anchor averaging.* In equation 23 we anchored at $m=1$; averaging the joint estimate over anchors ($m=1,\ldots,M$) lowers variance. (iii) *Bandwidths.* One can use modality-specific bandwidths $\tau_m$; the derivation is identical with $K_{\tau_m}$ per modality.

Define the (within-modality) kernel means $s_i^{(m)} \triangleq \frac{1}{B}\sum_{k=1}^B K_\tau(\mathbf{z}_i^{(m)}, \mathbf{z}_k^{(m)})$ and the (cross-modality-to-anchor) kernel means $c_i \triangleq \prod_{m=2}^M \frac{1}{B}\sum_{k=1}^B K_\tau(\mathbf{z}_i^{(1)}, \mathbf{z}_k^{(m)})$. Then the Hölder divergence estimator (up to an additive constant if using $\kappa$) is

$$
\widehat{D}_{\text{Hölder}} \;=\; \frac{1}{M}\sum_{m=1}^M \log\!\left(\frac{1}{B}\sum_{i=1}^B \big(s_i^{(m)}\big)^{M-1}\right) \;-\; \log\!\left(\frac{1}{B}\sum_{i=1}^B c_i\right).
\tag{24}
$$

All terms are differentiable; the computation costs $O(MB^2)$ and can be vectorized via kernel matrices $K_{ij}^{(m)} = K_\tau(\mathbf{z}_i^{(m)}, \mathbf{z}_j^{(m)})$ and $K_{ij}^{(m\to 1)} = K_\tau(\mathbf{z}_i^{(1)}, \mathbf{z}_j^{(m)})$.

Table 4: **Design space for uniformity and alignment**. Uniformity can be instantiated in Euclidean or manifold geometries; alignment can incorporate geometric constraints beyond pairwise distance.

| Principle | Space | Kernel/Metric | Notes |
|---|---|---|---|
| Uniformity (repulsion) | Euclidean ($\mathbb{R}^d$) | $\exp\!\left(-\|\mathbf{z}_i^{(m)} - \mathbf{z}_j^{(m)}\|_2^2/2\tau^2\right)$ | Gaussian kernel in $\mathbb{R}^d$; encourages spread. |
| | Unit Hypersphere ($\mathbb{S}^{d-1}$) | $\exp\!\left(-d_{\mathbb{S}}(\mathbf{z}_i^{(m)}, \mathbf{z}_j^{(m)})^2/2\tau^2\right)$ | Geodesic (Riemannian) Gaussian on $\mathbb{S}^{d-1}$. |
| Alignment (attraction) | Euclidean ($\mathbb{R}^d$) | $\|\mathbf{z}_i^{(m)} - \mathbf{z}_i^{(n)}\|_2^2$ | Pairwise matching per sample. |
| | Unit Hypersphere ($\mathbb{S}^{d-1}$) | $\left[d_{\mathbb{S}}(\mathbf{z}_i^{(m)}, \mathbf{z}_i^{(n)})\right]^2$ | Geodesic pairwise alignment. |
| | Area/volume preservation | $\sqrt{\det \mathbf{G}(\mathbf{z}^{(1)}, \dots, \mathbf{z}^{(M)})}$ | Penalizes global volume; $\mathbf{G}$ is the Gram Matrix. |

# D    DESIGN SPACE FOR UNIFORMITY AND ALIGNMENT

We illustrate some possible designs following our principle in Table 4, showing the generality and flexibility of our framework.

# E    COMPUTATIONAL COMPLEXITY

Our framework is formulated upon the core principles of feature alignment and uniformity, which are fundamental to the contrastive learning objective. The implementation of our method operates within the standard computational pipeline of modern contrastive learning. For a given batch of size $B$ with $d$-dimensional representations, the dominant computational cost remains the construction of the $B \times B$ pairwise similarity matrix, an operation with $O(B^2 d)$ time complexity. The objectives of alignment and uniformity are then computed based on this matrix, inheriting the same computational profile as the loss calculation and gradient backpropagation stages of standard contrastive methods.

Crucially, our formulation does not require any operations beyond those already present in the baseline. As such, our method introduces no additional computational overhead and shares an identical time and memory complexity profile with widely-used InfoNCE-based frameworks. This efficiency ensures our approach is scalable and readily applicable to large-scale training regimes.

# F    EXPERIMENTAL DETAILS

## F.1    IMPLEMENTATION DETAILS

All experiments use $4\times$ NVIDIA A6000 GPUs. We train with AdamW, a learning rate of $\mathbf{2 \times 10^{-5}}$, and a batch size of 128 per GPU (global batch $= 512$); other optimizer settings are default. For zero-shot T2V/V2T retrieval, we follow GRAM: each video clip is sampled with 8 frames during training, and the model is trained for 5 epochs. For cross-modal generation, we train on VGGSound for 50 epochs.

For hyperparameters, $\lambda_{\mathrm{uni}}$, $\lambda_{\mathrm{align}}$, and $\lambda_{\mathrm{vol}}$, we simply set them to 1 for retrieval, and use the temperature to control them. For generation, we use $\lambda_{\mathrm{vol}} = 0.1$ to have more emphasis on anchor-based alignment. Specifically, for the intra-modality uniformity and alignment terms, we use the temperature as $\tau = 0.07$, the same as the standard CLIP.

---

[1]If one uses the *unnormalized* kernel $\kappa(\mathbf{z}, \mathbf{z}') = \exp(-\|\mathbf{z} - \mathbf{z}'\|_2^2/(2\tau^2))$, then $\widehat{p}_m$ is scaled by a constant depending on $(d, \tau)$. This yields an *additive constant* in $D_{\mathrm{Hölder}}$ that does not affect optimization; we drop such constants in practice.

Table 5: Ablation on centroid uniformity temperature $\tau_{\text{ctr}}$ on MSR-VTT retrieval (Recall@1, %).

| $\tau_{\text{ctr}}$ | T2V R@1 | V2T R@1 | Avg R@1 |
|---|---|---|---|
| 0.01 | 56.0 | 54.4 | 55.2 |
| 0.03 | 56.8 | 53.2 | 55.0 |
| 0.07 | 57.4 | 53.2 | 55.3 |

## F.2 MORE EXPERIMENTAL RESULTS

### F.2.1 MORE ABLATION STUDIES

As we mentioned above, we use the temperature $\tau = 0.07$ for uniformity and the anchor-based alignment. However, a separate temperature $\tau_{\text{ctr}}$ for the volume-based centroid uniformity could control the global separability. Hence, we perform an ablation study on this parameter. Table 5 shows that the average performance is robust to the temperature $\tau$, while controlling centroid $\tau$ may affect the subtask performance (T2V and V2T).

### F.2.2 EXPERIMENTAL RESULTS ON MODALITY INTERPOLATION

Beyond the bimodal cases in Fig. 3, we present tri-modal interpolation results. Conditioning jointly on an image embedding, a text prompt, and an audio embedding, our model synthesizes images that integrate complementary semantics from all three modalities, demonstrating effective cross-modal fusion.

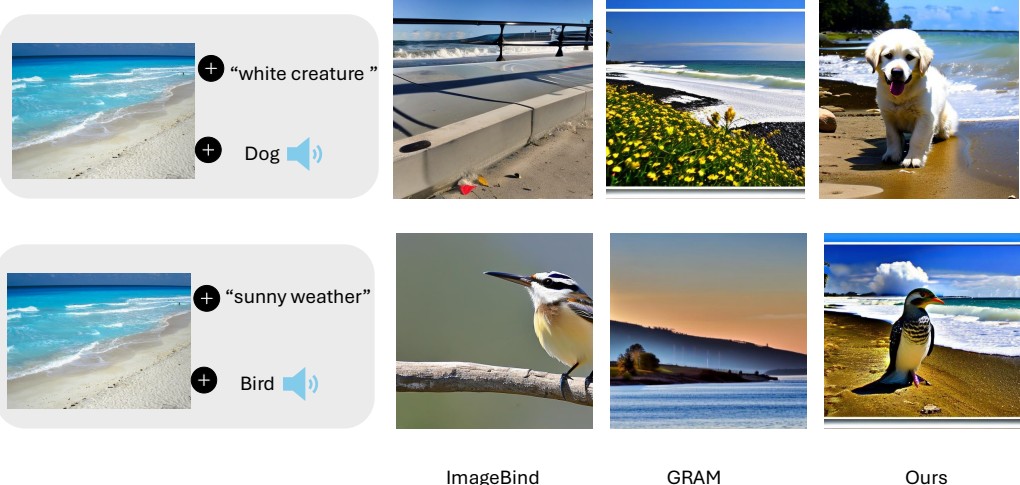

Figure 4: **Modality-interpolation generation results (V+T+A) → I.** When interpolating, vision, text, and audio representations, our method has a better ability to fuse the semantic information across modalities, leading to better generation.

## F.3 GENERATION RESULTS OF VGGSOUND

We present more generated samples from VGGSound in Fig. 5. Note that the image quality of VGGSound's videos is quite noisy, making the generation results similar. Also, we adopt a raw generation process for this demonstration, where the embeddings are directly passed to the decoder without additional conditions (e.g., negative prompts or quality-enhancing constraints). This can directly reflect the goodness of the multimodal alignment without external factors.

Audio to Image          Text to Image          Audio & Text to Image

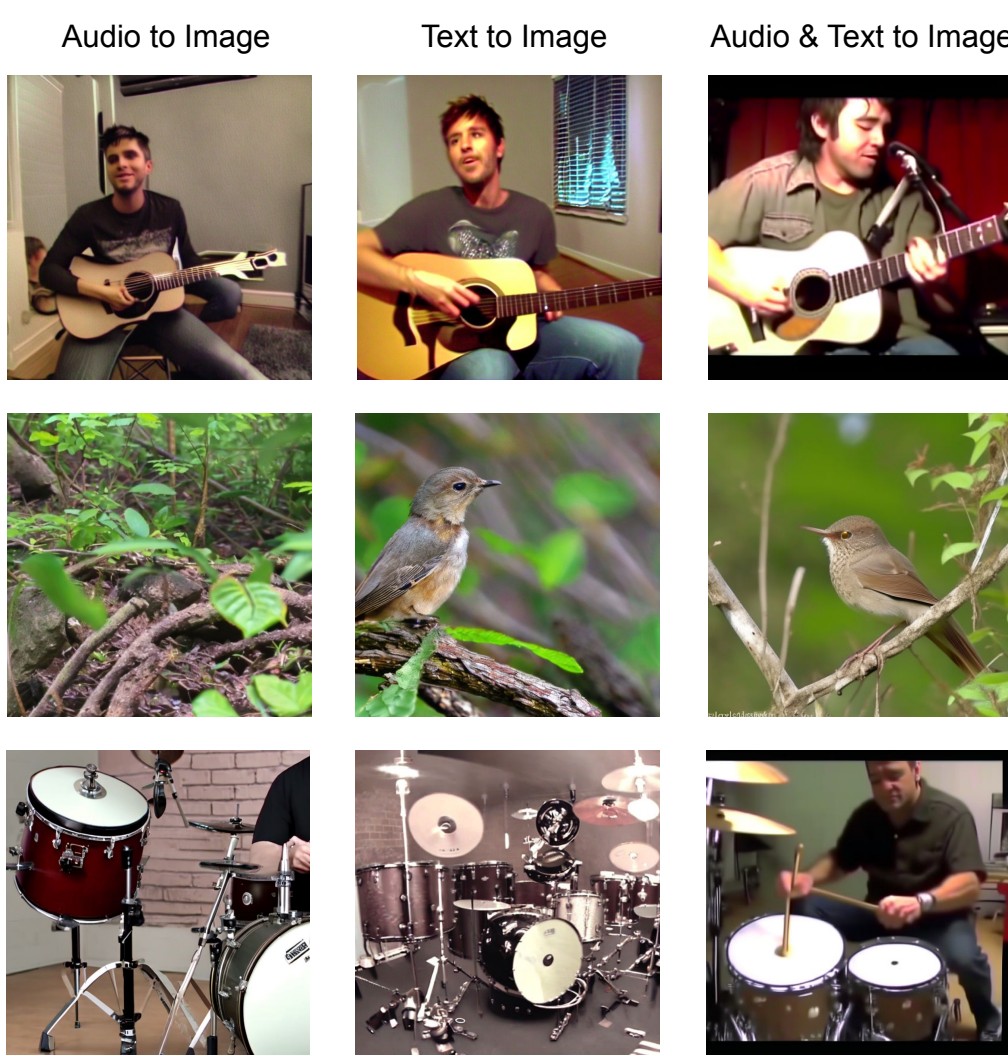

Figure 5: **More generated results from VGGsound.** We adopt a raw generation process for demonstrating the multimodal alignment ability, where the embeddings are directly passed to the decoder without additional conditions (e.g., negative prompts or quality-enhancing constraints)

