# OpenReview forum: "Towards Uniformity and Alignment for Multimodal Representation Learning"
_ICLR.cc/2026/Conference — ICLR 2026 Conference Withdrawn Submission_

### Official Review · Reviewer_ZwMh · 2025-10-16

**Soundness:** 4
**Presentation:** 2
**Contribution:** 3
**Rating:** 4
**Confidence:** 5

**Summary:**

The paper presents a novel model for multimodal learning that promotes alignment of matching pairs and uniformity over the hypersphere for more modalities through the volume loss from the GRAM paper.
Results are convincing and the idea is interesting as it improves some limitations of previous methods.
Also, the idea of testing the model in the generative task sounds and is convincing.

**Strengths:**

The idea sounds and wisely mixes two crucial topics that are scaling multimodal learning to mroe modalities and reduce the modality gap between modality representations.

I appreciate the colored summary box at page 4, as it helps fix the main concepts.

The results in the generation task are convincing and interesting.

Overall, the paper is interesting!

**Weaknesses:**

W1) In the box and throughout the paper, the authors say that it is crucial to have intra-modality uniformity and conflict-free alignment. Subsequently, they introduce the uniformity loss U(Z) and the L_align. However, they further add to the total loss the centroid uniformity, why? In this way, if I understood correctly, the total loss has two uniformity terms (U(Z) and (U(C)), plus the align term and the gram volume. I am afraid that the contribution of the uniformity losses may be too strong and disrupt the alignment, but the authors provide no further information for this.

- Also, why is U(C) necessary?

- I know that asking for ablation studies is somewhat boring and standard, but I am really curious to understand the contribution of each of the losses. Moreover, the authors provided simple ablations only on U(C) and L_vol, but a deeper understanding on the contribution of each of the loss is crucial.

- More theoretical explanations on the reason why adding two uniformity terms would be appreciated.

W2) How the proposed losses combination is justified by the theoretical analysis from the divergence perspective? If I'm not wrong (and I might be by the way), the divergence proves the claims regarding the uniformity and the alignment term, not for the combinations of the four losses proposed.

W3) (minor) The authors should revise the notation as the theoretical part of the paper is a bit heavy. Moreover, sometimes the modality index is between brackets and on top, sometimes is without brackets at the bottom. This makes the paper a bit hard to read and confuse the notation.

Minor comments:
- the references of LanguageBind and GRAM are wrong (still arxiv), please update them with the correct citation.
- line 131, InfoNEC instead of InfoNCE.

Overall, even though my initial score is not so high, I can lean towards increasing the score if the authors provide theoretical and/or empirical evidences in response to my comments.

**Questions:**

Q1) Can the authors provide ablation studies for each of the losses from scratch on MSRVTT?

Q2) Why the \lambda_vol for the generation is set to 0.1? the authors say that it is to have more emphasis on anchor-based alignment, but the volume definition is based on the anchor as well.

Q3) Can the authors compute the cosine similarity between true pairs to further strengthen the plot with the tsne?

Q4) can the authors share the link for their implementation? I would be curious to see losses implementation.

Q5) I assume that the results in Table 1 are achieved via the multimodal encoder that both VAST and GRAM employ. Can the authors provide retrieval results before feeding the embeddings to the multimodal encoder (which, in the case of both vast and gram is the text encoder)?

---

> ### Author Response · Authors · 2025-12-03
>
> Thank you for your insightful questions and suggestions. We have decided to resubmit this paper, but we would still like to address your main concerns, as we believe they are helpful for improving the manuscript.
>
> **1. Uniformity terms $U(Z)$ and $U(C)$ & additional ablations**
>
> We thank the reviewer for this thoughtful question.
> Based on your suggestion, we found that using only our core design, i.e., the uniformity term $U(Z)$ together with the alignment loss $L_{\text{align}}$, is comparable to learn strong representations and achieving good performance in the train-from-scratch setting. We train each experiment for 20 epochs, and report the cosine-similarity-based retrieval score. This new ablation is reported in the table below.
> | $U_C$ | $L_{\text{vol}}$ | T2VA | VA2T |
> |:-----:|:----------------:|:----:|:----:|
> | ✗     | ✗                | 17.8 | 25.6 |
> | ✗     | ✓                | 18.9 | 26.8 |
> | ✓     | ✗                | 19.2 | 27.1 |
> | ✓     | ✓                | 19.5 | 26.9 |
>
> This further supports our central claim that uniformity and alignment are the core of multimodal learning. The additional centroid uniformity and volume losses serve two purposes:
> Illustrating effective conflict-free variants.
>  The centroid uniformity $U(C)$ is motivated by the idea that a well-aligned multimodal tuple can be treated as a single “pseudo-unimodal” sample; increasing its uniformity improves separation between different tuples in the shared space. The volume loss $L_{\text{vol}}$ is an example of an alternative, conflict-free alignment design, which explicitly optimizes the collinearity of multimodal representations within each tuple.
>
>
> Further improving separability and performance. Empirically, we observe that increasing the separability of modality tuples via $U(C)$ leads to consistent performance gains, and adding the volume constraint $L_{\text{vol}}$ further improves results in our experiments.
>
> We will refine the presentation in the revision to more clearly highlight this principle and the roles of $U(Z)$, $L_{align}$, and present $U(C)$, and $L_{\text{vol}}$ as useful complementary variants. We appreciate the reviewer’s suggestion, which helped us both simplify the core design and clarify our analysis.
>
>
> **2. Connection to the divergence**
>
> We apologize for this confusion. Our divergence-based analysis is indeed focused on the core objective consisting of the conflict uniformity and alignment terms: these are the parts that we show can be interpreted as a practical kernel-based approximation of the proposed Hölder divergence, and thus are directly justified from the divergence perspective.
>
>
> **3. Notations, typos, and references.**
>
> Thank you for pointing out the issues of inconsistent notations, typos, and references. We will correct them in the revision.

---

### Official Review · Reviewer_BVa6 · 2025-10-29

**Soundness:** 2
**Presentation:** 1
**Contribution:** 2
**Rating:** 2
**Confidence:** 3

**Summary:**

This paper analyzes the inherent limitations of multimodal contrastive learning when using the InfoNCE objective. It identifies two sources of conflict that worsen as the number of modalities $M$ increases: the **alignment–uniformity conflict**, where uniformity forces oppose cross-modal attraction, and the **intra-alignment conflict**, caused by non-collinearity among multi-way positives. To address these issues, the authors propose **UniAlign**, which decouples intra-modality uniformity and cross-modality alignment via an anchor-based loss and a “volume” regularizer. They also introduce a new theoretical formulation — the *global Hölder divergence* — to interpret their objective as minimizing a cross-modal distribution gap. Experiments on retrieval and UnCLIP-style generation show modest but consistent improvements over GRAM and VAST baselines.

**Strengths:**

1. The paper tackles a relevant and well-known problem — the modality gap in multimodal contrastive learning — and extends previous analyses from the bimodal to the general multimodal case.
2. The proposed framework is straightforward to implement, requiring no architectural modifications or additional modules.
3. The empirical results are encouraging, showing that the proposed method consistently outperforms existing alternatives.

**Weaknesses:**

## Major
1. **Theoretical clarity.** The theoretical section is difficult to follow, and several key definitions are vague or insufficiently motivated. In particular:
   - In Eq. (3), it is unclear why $V_a$ represents the alignment force and $\Phi_a$ the uniformity force — this connection should be explained more carefully.
   - Assumption 1 is introduced without justification; it is not obvious why it should hold or be meaningful in practice.
   - In the boxed text of Section 3.1, the statement *“promote uniform coverage within that modality only”* is ambiguous and should be clarified.
   - The same box refers to *“consensus magnitude”* as if it were a standard concept, but it is never defined or explained.

2. **Idealized assumptions.** The theoretical analysis relies on simplified and somewhat unrealistic conditions (e.g., independence and isotropy of negatives, uniform temperature across modalities). A discussion of the robustness of the results under more practical conditions would improve the paper.

3. **Loose theory–practice connection.** The derivation of the Hölder divergence (Eq. 16) is mathematically elegant but remains disconnected from the practical loss in Eq. (13). The claim that UniAlign effectively minimizes this divergence is heuristic, relying on a kernel-density approximation that is never empirically verified.

## Minor
1. Corollaries should follow from a theorem or proposition. Please revise the naming (e.g., use *Theorem*, *Lemma*, or *Proposition*).
2. The method introduces several hyperparameters whose influence is not systematically analyzed. An ablation or sensitivity study would strengthen the experimental section.

**Questions:**

1. Could similar improvements be obtained by simply re-weighting InfoNCE gradients rather than introducing new loss terms?
2. Can you empirically estimate the proposed Hölder divergence during training to verify that it indeed decreases?

---

> ### Author Response · Authors · 2025-12-03
>
> **1. 1 Eq. 3 clarification.**
>
> Eq. (3) is obtained by explicitly decomposing the gradient of the InfoNCE loss w.r.t. the embedding of sample (i) in modality (a). The first vector only depends on the positive pair ((i,i)) across modalities and consistently pulls the paired embeddings closer (increasing their cosine similarity). This is exactly the “alignment” component in standard analyses of contrastive learning, e.g., it is the only remaining term if we keep just the positive logit in InfoNCE.
> The second vector is a (softmax-weighted) sum over all other samples (j\neq i) and acts as a repulsive force that pushes the embedding away from negatives. On the unit sphere, this repulsion can be interpreted as pushing the empirical density toward a more spread-out (approximately uniform) configuration, as commonly discussed in prior work on contrastive learning and modality gap. For this reason, we refer to this part as the “uniformity” force. We will add a short explanation around Eq. (3) to make this decomposition and terminology precise.
>
> **1.2 & 2. Assumption 1 clarification**
>
> We apologize for the confusion. Assumption 1 is a mild geometric assumption that we explicitly motivate both in the main text and in Appendix A.
>
> Assumption 1 is meaningful both empirically and mathematically. Empirically, it reflects a standard and widely observed property of contrastive learning: **hard negatives** (semantically similar but distinct samples) tend to lie in directions close to the positive pair, while the bulk of **easy negatives** is more isotropic and behaves like zero-mean noise [1]. The isotropic assumption is a mild and meaningful approximation in our setting. In large-batch contrastive learning with diverse data, the vast majority of negatives for a given anchor are semantically unrelated and spread over many directions in the embedding space. On the unit hypersphere, such “background” negatives behave like almost random directions whose mean is close to zero and whose covariance is close to a multiple of the identity.
>
> In Eq. (3), this is exactly captured by decomposing the uniformity force from each non-anchor modality $n \neq a$ as
> $\Phi^{(n)}_a = c_n \hat V_a + \varepsilon_n,$
> where the coefficients $c_n \ge c_0 > 0$ model the systematic contribution of hard negatives along the alignment direction $\hat V_a$, and the residuals $\varepsilon_n$ represent dispersed easy negatives with approximately zero mean and bounded covariance. This matches the typical hard–easy negative geometry discussed in prior analyses of InfoNCE.
>
> From a theoretical perspective, we need a mild structural assumption of this form that allows us to rigorously separate a **coherent component** (which accumulates linearly with the number of modalities $M$) from **noise-like components** (which largely cancel by averaging). This is what enables Corollary 1: we can formally show that the alignment–uniformity conflict grows with $M$, instead of merely arguing heuristically. Without such an assumption, we would only obtain very loose worst-case bounds that do not reflect the geometry actually seen in modern contrastive models. We will make this role of Assumption 1 clearer in the revision.
>
>
> [1]. Fang X, Li J, Sun Q, et al. Rethinking the uniformity metric in self-supervised learning[J]. arXiv preprint arXiv:2403.00642, 2024.
>
>
> **1.3 Statement “promote uniform coverage within that modality only”.**
>
> By “promote uniform coverage within that modality only”, we mean that the uniformity term is applied separately to the embeddings of each modality, i.e., it encourages the anchor-modality embeddings ${z_i^{(a)}}$ to be well-spread on the unit sphere (in the standard “uniformity on the hypersphere” sense), without enforcing any cross-modality uniformity.
>
> **1.4 “Consensus magnitude” clarification.**
>
> The term “consensus magnitude” means measuring **how well the modalities within a tuple agree in direction**. We will clarify this in the revision.
>
>
> **3. & 7 Theory-practice connection &  Hölder divergence during training.**
>
> Our training loss can be viewed as an empirical, tractable estimator of the Hölder divergence between modality distributions. In the revision, we make this connection more explicit and empirically estimate the Hölder divergence during training. We provide a new training curve showing that this estimated divergence consistently decreases as UniAlign trains, confirming that our practical loss indeed drives down the proposed Hölder divergence.
>
>
> **4. Naming for Corollaries.**
>
> Thank you for the suggestion. We revise the name to Proposition.
>
>
> **5. re-weighting InfoNCE**
>
> Simply re-weighting InfoNCE cannot replicate our improvements. Re-weighting (e.g., via a global coefficient or temperature) only rescales the magnitude of the same entangled gradient, but does not change the directional conflict between alignment and cross-modality uniformity that we analyze in Eq. (3).

---

### Official Review · Reviewer_EfDF · 2025-10-30

**Soundness:** 3
**Presentation:** 3
**Contribution:** 2
**Rating:** 6
**Confidence:** 2

**Summary:**

This paper identifies two fundamental conflicts that hinder multimodal contrastive learning, the alignment–uniformity conflict and the intra-alignment conflict, and proposes a principled decoupling framework with a theoretical guarantee via a global Hölder divergence to achieve conflict free multimodal representation learning that improves both discriminative and generative performance.

**Strengths:**

1.The paper is clearly written, presenting the motivation, conflicts, and proposed solution in a well-structured manner.

2..The proposed method is theoretically grounded and experimentally validated, demonstrating robust performance across multimodal retrieval and generation tasks.

**Weaknesses:**

1.There are some writing errors, such as “InfoNEC” in Section 2.2.
2.Although anchor-based alignment eliminates cross-modal rejection, it introduces modal bias. I have a question: Could the selection of different anchor modalities lead to representation imbalance?
3.Your approach employs intra-modal consistency to prevent representation collapse. Was modal collapse considered during training?
4.The proposed global Hölder divergence is defined over multiple modality distributions. Is this divergence sensitive to the curse of dimensionality in high-dimensional embedding spaces?

**Questions:**

Specific issues can be found in the weaknesses.

---

> ### Author Response · Authors · 2025-12-03
>
> **1. Typos.**
>
> Thank you for pointing out the types. We have revised them.
>
> **2. Could the selection of different anchor modalities lead to representation imbalance?**
>
> We do not expect this to cause an imbalance. Our objective minimizes pairwise distances between modalities, so the shared representation is determined by these cross-modal distances, not by which modality is chosen as “anchor.” Changing the anchor corresponds to a different parametrization of the same joint space rather than biasing the representations.
>
>
> **3. Modal collapse.**
>
> Yes. Without a proper uniformity component, the optimization would be dominated by alignment and could drive all modalities toward a collapsed solution. Our intra-modal consistency and anchor-only uniformity terms are introduced exactly to avoid this: they keep each modality well-spread while still aligning them, and in practice, we do not observe representation collapse during training.
>
> **4. Sensitive of Hölder divergence to the high-dimension.**
>
> We would like to first clarify that we do not directly optimize the global Hölder divergence during training. Instead, our loss is a practical kernel-based approximation of this divergence, and the Hölder formulation is used to theoretically explain why our objective reduces the distribution discrepancy among multiple modalities.
> Sensitivity to high-dimensionality mainly comes from the KDE approximation rather than from the Hölder divergence itself. Directly minimizing the KDE-based Hölder divergence as a standalone loss did not train well; however, when we use our proposed objective, our empirical estimate of the Hölder divergence consistently decreases over training. This indicates that, despite potential high-dimensional issues, our practical formulation still effectively drives down the intended divergence.

---

### Official Review · Reviewer_WFru · 2025-10-31

**Soundness:** 2
**Presentation:** 2
**Contribution:** 2
**Rating:** 2
**Confidence:** 4

**Summary:**

The authors claim that standard contrastive learning over more than two modalities is limited by a conflict between the alignment and uniformity criteria. To this end, they propose an approach, UniAlign, which aims to resolve this conflict. They theoretically analyze the identified form of conflict and empirically evaluate their approach on generative and retrieval tasks.

**Strengths:**

1. The work addresses a potential roadblock in contrastive learning, i.e., the possibility that the alignment of positive tuples being interfered with by the uniformity criterion in contrastive learning.
2. The authors do a good job in conveying the intuition behind the problem that they are tackling.
3. For the kind of alignment-uniformity conflict presented in their Assumption 1, the authors theoretically prove that it increases as the number of modalities grow.

**Weaknesses:**

1. Although the authors argue for the existence of a cross-modality uniformity conflict which oppose alignment, there is no clear evidence for this, either in the existing literature or in this work.
For instance, in Line 104, the authors refer to learning, Yin et al. (2025), mentioning that "clearly demonstrate that uniformity across modalities (“inter-uniformity”) conflicts with the alignment term". However, I found no results as such upon going through that work. The claim can be described as: for each positive pair, the alignment term can be cancelled out by the uniformity forces when aggregated across modalities. However, at least intuitively, the probability of this happening seems extremely low, because (i) otherwise most multimodal contrastive methods just would not work; and (ii) it is well known that there exists clusters of similar positive pairs (even in the unsupervised case) which are spontaneously grouped together under contrastive learning [a, b], meaning that the probability that alignment of positive pairs would be reinforced by other similar positive pairs is much higher than the probability of interference from unrelated / negative pairs. Having said that, it is possible that the cancellations could get stronger as the number of modalities increase as the authors argue, however, to establish this fact there needs to be significantly more theoretical and empirical analyses, since there is plenty of very strong evidence pointing to the contrary [c, d].

2. No ablation studies are presented in this paper. Without them, it is difficult to evaluate exactly what is contributing to the performance improvements reported in Table 1.

3. The purpose of the volume-based complement loss is not clear. It seems that it encourages samples that are grouped together from multiple modalities into a tuple a dispersed. However, this would mean that samples that share the same semantic information would be pushed apart, which goes against the desired objective.

4. The L_align term does not seem to be any different from a standard contrastive learning objective, where one modality is considered as an anchor. However, there does seem to be a downside, which comes from imposing only the uniformity objective on the anchor modality. It would imply that samples that are semantically similar in the anchor modality will not be brought together, and consequently, neither would the samples from the other modalities, since such is the nature of the anchor to which they are aligned.

References:

[a] Parulekar et al., "InfoNCE Loss Provably Learns Cluster-Preserving Representations", COLT 2023. \
[b] Lu et al., "f-MICL: Understanding and Generalizing InfoNCE-based Contrastive Learning", TMLR 2023. \
[c] Girdhar et al., "IMAGEBIND: One Embedding Space To Bind Them All", CVPR 2023. \
[d] Wang et al., "Image as a Foreign Language: BEIT Pretraining forVision and Vision-Language Tasks", CVPR 2023.

**Questions:**

Please refer to the Weaknesses section.

---

> ### Author Response · Authors · 2025-12-03
>
> **1. Theoretical and empirical supports for the uniformity and alignment conflict.**
>
> We would like to clarify that the uniformity-alignment conflict exists in **multimodal learning**, rather than unimodal learning (e.g., standard contrastive learning). The modality gap caused by InfoNCE has been discovered and analyzed in many previous works. Liang et al. 2022 clearly mentioned that InfoNCE in CLIP may lead to a larger modality gap, while Shi et al., 2023 intuitively described the uniformity-alignment conflict in vision-language. Hence, there exists sufficient evidence in the existing literature (more references: [1, 2]) to support the observation of modality gap caused by InfoNCE.
>
> Yin et al. (2025) give a mathematical explanation of the uniformity-alignment conflict. Both our analysis and prior works show that, under InfoNCE, the uniformity component can systematically oppose the alignment gradient across modalities. Importantly, we do not claim that alignment is fully cancelled so that contrastive learning “would not work”, but that the net alignment signal is weakened by this opposing force, especially as the number of modalities grows.
>
> We will clarify this in the revision.
>
> Regarding the two reasons based on [a,b]: both works study InfoNCE in **single-modality** settings, where samples come from a single distribution
> $x \sim p(x)$, and positives are generated by augmentations $(x, x^+)$ with $x^+ \sim q(\cdot \mid x).$ The cluster-preservation results they prove are about **intra-modality** geometry and do not address how different modalities (e.g., image vs. text vs. audio) interact through shared negatives. Mathematically, our setting is different: we consider pairs
> $(x, y) \sim p(x, y), \quad x \in \mathcal{X}, y \in \mathcal{Y},$ with separate encoders for each modality, which introduce additional **cross-modality terms** in the InfoNCE gradient decomposition that are absent in [a,b].
>
>
> Moreover, ImageBind [c] explicitly reports strong sensitivity to the temperature, which is consistent with our view that temperature controls the uniformity–alignment trade-off rather than contradicting it. Empirically, Fig. 2 and our UnCLIP-style generation experiments (Table 3) show that standard InfoNCE-trained models (ImageBind) exhibit clear modality clusters and significantly worse cross-modal FID than our method, directly evidencing a modality gap that our decoupled objective mitigates. Regarding [d], we also do not view [d] as evidence against the existence of a cross-modality uniformity–alignment conflict. Wang et al. treat images “as a foreign language” and pre-train BEiT with a mixture of objectives (masked image/language modeling, translation-style tasks, etc.) on top of a shared multimodal backbone, rather than a pure CLIP-style InfoNCE loss.
>
>
>
> [1]. Wang Z, Zhao Y, Huang H, et al. Connecting multi-modal contrastive representations[J]. Advances in Neural Information Processing Systems, 2023, 36: 22099-22114.
>
> [2]. Mistretta M, Baldrati A, Agnolucci L, et al. Cross the gap: Exposing the intra-modal misalignment in clip via modality inversion[J]. ICLR, 2025.
>
>
> **2. Ablation Study.**
>
> The ablation study on each component is provided in Table 2 in the main manuscript.
> We additionally provide the train-from-scratch ablation on MSRVTT. It shows that both the volume-based uniformity and alignment can effectively increase embedding separability, and thus improve the retrieval performance on MSRVTT.
>
> | $U_C$ | $L_{\text{vol}}$ | T2VA | VA2T |
> |:-----:|:----------------:|:----:|:----:|
> | ✗     | ✗                | 17.8 | 25.6 |
> | ✗     | ✓                | 18.9 | 26.8 |
> | ✓     | ✗                | 19.2 | 27.1 |
> | ✓     | ✓                | 19.5 | 26.9 |
>
>
> **3. Purpose of volume-based loss.**
>
> The volume-based complement loss does not push semantically similar tuples apart. It acts within each tuple: it shrinks the simplex spanned by the modality embeddings so that modalities of the same sample become more collinear, implementing our principle of reducing alignment conflict via cross-modal collinearity. Separating different tuples and hard negatives is handled by the uniformity term.
>
> **4. Difference from contrastive learning objective.**
>
> Our $L_{\text{align}}$ is intentionally simpler than a standard contrastive loss: it is a pure positive-pair term (no negatives), while contrastive/uniformity behaviour is handled separately by the anchor’s uniformity loss. This decoupling reduces multimodal gradient conflicts: the anchor serves as a uniform template, and other modalities align to it along a single direction. Uniformity on the anchor prevents collapse but does not forbid local clusters, and this structure is propagated to other modalities via $L_{\text{align}}$. Even in standard CLIP/InfoNCE, semantically similar unlabeled samples are often treated as hard negatives and are repelled rather than explicitly pulled together.

---

### Author Response · Authors · 2025-12-03

Dear ACs and reviewers,

We sincerely thank you for the time and effort you have dedicated to our manuscript. We have decided to resubmit this work in the future.

In the meantime, we would like to respond to the major concerns raised by each reviewer in order to clarify several misunderstandings and questions, and to keep the discussion open for the community. We are especially grateful to reviewer EfDF for recognizing our paper and reviewer ZwMh for identifying it as interesting, as well as for the insightful suggestions from all reviewers.


Best regards,

The Authors

---

### Note · Authors · 2025-12-11

I have read and agree with the venue's withdrawal policy on behalf of myself and my co-authors.